# A Clinically Significant Prostate Cancer Predictive Model Using Digital Rectal Examination Prostate Volume Category to Stratify Initial Prostate Cancer Suspicion and Reduce Magnetic Resonance Imaging Demand

**DOI:** 10.3390/cancers14205100

**Published:** 2022-10-18

**Authors:** Juan Morote, Ángel Borque-Fernando, Marina Triquell, Miriam Campistol, Anna Celma, Lucas Regis, José M. Abascal, Pol Servian, Jacques Planas, Olga Mendez, Luis M. Esteban, Enrique Trilla

**Affiliations:** 1Department of Urology and Surgery, Vall d’Hebron Hospital and Universitat Autònoma de Barcelona, 08035 Barcelona, Spain; mtriquell@vhebron.net (M.T.); mcampistol@vhebron.net (M.C.); acelma@vhebron.net (A.C.); lregis@vhebron.net (L.R.); jplanas@vhebron.net (J.P.); etrilla@vhebron.net (E.T.); 2Department of Urology, Hospital Miguel Servet, IIS-Aragon, 50009 Zaragoza, Spain; aborque@comz.org; 3Departments of Urology and Surgery, Parc de Salut Mar and Universitat Pompeu Fabra, 08003 Barcelona, Spain; jabascal@psmar.cat; 4Department of Urology, Hospital Germans Trias I Pujol, 08916 Badalona, Spain; pservian.germanstrias@gencat.cat; 5Urology Biomedical Research Unit, Vall d´Hebron Research Institute, 08035 Barcelona, Spain; olga.mendez@vhir.org; 6Department of Applied Mathematics, Escuela Universitaria Politécnica La Almunia, Universidad de Zaragoza, 50100 Zaragoza, Spain; lmeste@unizar.es

**Keywords:** prostate cancer, suspicion, clinically significant, predictive model, risk calculator, external validation, magnetic resonance imaging, development, external validation

## Abstract

**Simple Summary:**

Early detection of PCa (PCa) has evolved towards clinically significant PCa (csPCa) after the spread of pre-biopsy multiparametric magnetic resonance imaging (mpMRI). However, PCa suspicion remains based on prostate-specific antigen (PSA) elevation and/or abnormal digital rectal examination (DRE). This change of paradigm and approach has reduced unnecessary prostate biopsies and overdetection of insignificant PCa, while the demand for mpMRI has skyrocketed despite its implementation not being allowed at all sites. The European Association of Urology (EAU) proposes risk-organized models for early detection of csPCa stratifying the initial PCa suspicion to reduce MRI scans and then prostate biopsies after mpMRI. Risk calculators are efficient tools for individualizing the risk of csPCa, especially when prostate volume is included in the predictive models. After the development and external validation of the Barcelona MRI risk calculator (BCN-RC 2) for the selection of candidates for prostate biopsy, we have now developed and externally validated BCN-RC 1 with the aim of reduce mpMRI demand. Both BCN-RC 1 and RC 2 are ready to be integrated in a risk-organized model for early detection of csPCa.

**Abstract:**

A predictive model including age, PCa family history, biopsy status (initial vs repeat), DRE (normal vs abnormal), serum prostate-specific antigen (PSA), and DRE prostate volume ca-tegory was developed to stratify initial PCa suspicion in 1486 men with PSA > 3 ng/mL and/or abnormal DRE, in whom mpMRI followed; 2- to 4-core TRUS-guided biopsies where Prostate Imaging Report and Data System (PI-RADS) > 3 lesions and/or 12-core TRUS systematic biopsies were performed in one academic institution between 1 January 2016–31 December 2019. The csPCa detection rate, defined as International Society of Uro-Pathology grade group 2 or higher, was 36.9%. An external validation of designed BCN-RC 1 was carried out on 946 men from two other institutions in the same metropolitan area, using the same criteria of PCa suspicion and diagnostic approach, yielded a csPCa detection rate of 40.8%. The areas under the receiver operating characteristic curves of BCN-RC 1 were 0.823 (95% CI: 0.800–0.846) in the development cohort and 0.837 (95% CI: 0.811–0.863) in the validation cohort (*p* = 0.447). In both cohorts, BCN-RC 1 exhibited net benefit over performing mpMRI in all men from 8 and 12% risk thresholds, respectively. At 0.95 sensitivity of csPCa, the specificities of BCN-RC 1 were 0.24 (95% CI: 0.22–0.26) in the development cohort and 0.34 (95% CI: 0.31–0.37) in the validation cohort (*p* < 0.001). The percentages of avoided mpMRI scans were 17.2% in the development cohort and 22.3% in the validation cohort, missing between 1.8% and 2% of csPCa among men at risk of PCa. In summary, BCN-RC 1 can stratify initial PCa suspicion, reducing the demand of mpMRI, with an acceptable loss of csPCa.

## 1. Introduction

The European Randomised Screening Prostate Cancer (ERSPC) trial continues to show that early detection of clinically significant prostate cancer (csPCa) decreases PCa mortality beyond twenty years of follow-up [1]. After the stated position of the US Preventive Services Task Force against PCa screening with prostate-specific antigen (PSA), due to the high percentage of unnecessary prostate biopsies and overdetection of insignificant PCa (iPCa) [2], early detection of PCa has evolved towards csPCa due to the spread of pre-biopsy multiparametric magnetic resonance imaging (mpMRI) [3]. However, PCa suspicion remains based on serum PSA elevation and/or abnormal digital rectal examination [4,5]. Currently, mpMRI exhibits a high enough negative predictive value to allow for the avoidance of prostate biopsies when the Prostate Imaging Report and Data System (PI-RADS) score is less than 3 [6]. In addition, mpMRI also allows targeted biopsies on suspicious lesions, improving the sensitivity for csPCa of classic systematic prostate biopsies [7,8]. Therefore, the current approach of csPCa requires pre-biopsy mpMRI after serum PSA > 3.0 ng/mL and/or abnormal DRE and targeted biopsies to Prostate Imaging Report and Data System (PI-RADS) lesions ≥ 3, complemented with systematic prostate biopsies [4]. This paradigm change in the early detection of PCa has triggered the demand for mpMRI scans, which are not allowed at some sites [9]. In experienced centers, mpMRI has been replaced with biparametric MRI (bpMRI), reducing the MRI scan time fourfold while maintaining the quality and reproducibility of PI-RADS [10]. Further, PSA density, modern markers, and predictive models have been proposed to improve the selection of candidates for prostate biopsy after MRI in uncertain scenarios where a high percentage of unnecessary biopsies or overdetection of iPCa remain [11]. However, any increase in the current cost of early detection of csPCa is generally rejected [12]. The European Association of Urology (EAU) proposes risk-organized models of early detection of csPCa based on stratifying the initial PCa suspicion to reduce unnecessary MRI scans and, after that, a new stratification for reducing unnecessary prostate biopsies [9,13].

Prostate volume is a valuable predictor of csPCa, its inverse relationship with csPCa risk having recently been confirmed in a systematic review of studies carried out over the last thirty years [14]. An accurate prostate volume is essential for PSA density calculation, requiring measurement by transrectal ultrasonography (TRUS) since its introduction [15]. Today, MRI provides the most accurate prostate volume measurement [16]; TRUS is mainly used to perform prostate biopsies but infrequently to assess prostate volume alone [17]. However, prostate volume is needed for PSA density calculation and for some new markers assessment [18] to improve the efficacy of predictive models to stratify the initial suspicion of PCa [9]. Roobol et al. showed how the Rotterdam risk calculator´s prostate volume estimated with DRE can replace TRUS prostate volume [19,20]. Classically, urologists have performed a complete DRE, with great efficiency, detecting abnormalities in the posterior prostate gland surface and categorizing the prostate volume [21]. Regrettably, the use of DRE is limited even among physicians who diagnose and treat prostate cancer [22].

After developing and externally validating the Barcelona MRI predictive model and designing the risk calculator, now named BCN-RC 2, for selecting proper candidates for prostate biopsy after mpMRI [23], we aim to develop a new predictive model of csPCa to stratify the initial suspicion of PCa, including the DRE prostate volume category, for avoiding unnecessary MRI scans. External validation of designed web BCN-RC 1 in the Barcelona metropolitan area was also performed. Our second objective is to propose a risk-organized model for the early detection of csPCa by sequencing both BCN-RC 1 and RC 2.

## 2. Materials and Methods

This predictive model was developed and externally validated from the same cohorts in which BCN-RC 2 was developed and externally validated.

### 2.1. Development Cohort

A group of 1486 men with serum PSA > 3 ng/mL and/or abnormal DRE were recruited from 1 January 2016–31 December 2019 in an academic institution of the metropolitan area of Barcelona (Vall d´Hebron Hospital), Spain. Pre-biopsy 3 Tesla mpMRI was performed followed by 2- to 4-core transrectal TRUS visual-guided biopsies for all PI-RADS v.2 >3 lesions and 12-core TRUS systematic biopsy and 12-core TRUS systematic biopsy in men with PI-RADS v.2 < 3. The dataset was recruited prospectively according to the standards of reporting for MRI-targeted biopsy studies (START) [24]. Men under 5-α reductase inhibitors and prior PCa detected were excluded, as well as those with prior atypical small acinar proliferation or high-grade prostate intraepithelial neoplasia with atypia.

### 2.2. Validation Cohort

The validation was formed with 946 men with the same criteria of PCa suspicion as those of the development cohort, retrospectively recruited in two academic institutions (Parc de Salut Mar and Germans Trias i Pujol Hospital) of the Barcelona metropolitan area, Spain. The diagnostic approach of csPCa was also the same as that of the development cohort.

### 2.3. MpMRI Characteristics

MpMRI was acquired with 3-Tesla scanners and surface phased-array coil. T2-weighted imaging (T2W), diffusion-weighted imaging (DWI), and dynamic-contrast-enhanced (DCE) imaging were analyzed according to the guidelines of European Society of Urogenital Radiology [25]. An expert radiologist with more than three years’ experience and more than 300 reports per year reported mpMRI scans with the PI-RADS v.2.0 [26]. All reports of mpMRI included the prostate volume.

### 2.4. DRE-Prostate Volume Category Assessment

The prostate volume categories assessable with DRE were defined as category I when MRI prostate volume was less than 30 mL, category II when it was between 30 and 59 mL, and category III when MRI prostate volume was 60 mL or higher [19,20].

### 2.5. CsPCa Definition

All cores were separately submitted to the pathology departments. Expert pathologists analyzed material and csPCa was defined by the International Society of Uro-Pathology (ISUP) grade group 2 or higher [27].

### 2.6. Predictive Model Development

Age (years), ethnicity (Caucasian vs non-Caucasian), serum PSA level (ng/mL), DRE (normal vs abnormal), PCa family history (no vs first-degree), biopsy status (biopsy-naïve vs prior negative biopsy), and DRE prostate volume category (I to III) were explored as predictive variables of csPCa.

### 2.7. Endpoint Measurements

Avoided mpMRI scans and missed csPCa.

### 2.8. Statistical Analysis

Reporting tumour marker prognostic studies recommendations were followed (REMARK) [28], as well as update standards for reporting diagnostic accuracy studies (STARD 2015) [24]. Comparison between proportions and medians were made with the chi-squared test and Mann–Whitney U tests. A binary logistic regression of csPCa candidate predictors was used to generate the predictive model. Continuous variables were modeled as linear or nonlinear predictors using restricted cubic splines. Calibration of the predictive model was assessed in developed and validation cohorts. Discrimination ability was assessed from receiver operating characteristic (ROC) curves [29], and areas under the curve (AUC) were compared with the DeLong test [30]. The net benefit of stratifying PCa suspicion with the predictive model and perform mpMRI to all men was analyzed with decision curve analysis (DCA) [31]. Clinical utility was assessed via clinical utility curve (CUC), exploring potential rates of missed csPCa and avoided mpMRI scans [32]. Specificities for the 80, 85, 90, and 95% sensitivity of csPCa of the predictive model in development and external validation cohorts were compared with chi-squared test. Sensitivity, specificity, and positive and negative predictive value were analyzed, and the rates of avoidable MRI exams and potentially undetected csPCa were analyzed. Odds ratios (OR) and 95% confidence intervals (CI) were calculated. For external validation, transparent reporting of the multivariable prediction model for individual prognosis or diagnosis (TRIPOD) statements followed. Statistical analysis was conducted with the R programming language v.4.0.3 (R Foundation for Statistical Computing, Vienna, Austria) and SPSS v.25 (IBM, statistical package for social sciences, San Francisco, CA, USA).

## 3. Results

### 3.1. Characteristics of the Development and Validation Cohorts

Characteristics of the development and validation cohorts are summarized in Table 1. We note significance in younger men with higher serum PSA in the external validation cohort (*p* < 0.001). We also note a higher percentage of abnormal DRE and repeat prostate biopsies in the validation cohort, compared with a lower percentage of men with PCa family history (*p* < 0.001). Prostate volume was similar in both cohorts (*p* = 0.559). Caucasian ethnicity was predominant in both cohorts (*p* = 0.738). A similar distribution of DRE prostate volume categories was observed in both cohorts (*p* = 0.675). There was a different case-mix of PI-RADS categories in both cohorts (*p* < 0.001). An increased trend of csPCa and a significant increase of iPCa (*p* < 0.001) was found in the validation cohort (*p* = 0.058). The distribution of csPCa according to the PI-RADS categories was similar in both cohorts when they were > 3, and a higher percentage of csPCa existed in men with PI-RADS < 3 in the validation cohort (*p* < 0.001).

### 3.2. Development of the Predictive Model and Its Calibration in the Development and Validation Cohorts

Independent predictors of csPCa were selected to generate the predictive model; ORs with 95% CIs in univariate and multivariate analysis are presented in Table 2. We note that the DRE prostate volume category was an independent predictor of csPCa. The non-linear dependence of PSA analysed by restricted cubic splines was adjusted by a logarithmic transformation. The developed nomogram is presented in Figure 1. Calibration curves of the developed model in the development and validation cohorts are presented in Figure 2. We note a good agreement between predictions and real outcomes in both development (A) and validation (B) cohorts. In the validation cohort, a slight underestimation of csPCa incidence was observed with a calibration of approximately 0.118, showing a minimum difference between the mean observed and the mean predicted and an almost perfect slope of 1.013.

### 3.3. Discrimination Ability of BCN-RC 1 for csPCa, Net Benefit over Performing mpMRI in All Men, Clinical Utility and Performance in the Development and Validation Cohorts

A web app for BCN-RC 1 was designed and is freely available with the BCN-RC 2 at https://mripcaprediction.shinyapps.io/MRIPCaPrediction/ (accessed on 16 October 2022). The discrimination ability for csPCa of BCN-RC 1 in the development and validation cohorts, analysed through ROC curves, are presented in Figure 3. The AUC of BCN-RC 1 was 0.823 (95% CI: 0.800–0.846) in the development cohort and 0.837 (95% CI: 0.810–0.863) in the validation cohort (*p* = 0.447).

BCN-RC 1 showed net benefit, analysed with DCAs, over performing mpMRI in all men in the development cohort from 12% threshold probability (A) and from 8% threshold probability in the validation cohort (B), Figure 4.

Finally, the clinical utility of BCN-RC 1 in the development and validated cohorts is reported through CUCs in Figure 5. Percentages of avoided mpMRI and missed csPCa are presented for a continuous expression of csPCa risk threshold.

Specificities obtained with the probability thresholds at 0.80 to 0.95% sensitivities for csPCa are presented in Table 3. We note that at 11.1% and 13.3%, which were the 0.95 sensitivity thresholds, that specificity in the development cohort was 0.24 (95% CI: 0.22–0.26), compared with 0.34 (95% CI: 0.31–0.37) in the validation cohort (*p* < 0.001), Table 3. At 0.80 and 0.85 sensitivities, the specificities were similar in both development and validation cohorts, while at 0.90 and 0.95 sensitivities, the specificities in the validation cohort were significantly higher than those observed in the development cohort.

We found that, using the probability threshold of missing 5% of all detected csPCa, which represents 1.8% and 2% of missed csPCa among the men at risk of PCa in the development and validation cohorts, respectively, the percentage of saved mpMRI scans was 17.2% in the development cohort but reached up to 22.3% in the validation cohort (*p* < 0.001) (Table 4, Figure 5). Performance parameters of BCN-RC 1 at 0.95 sensitivity threshold (11.1%) in the development cohort and validation cohort (13.3%) are presented in Table 4.

Finally, Table 5 describes the number of missed csPCa and saved mpMRI scans in a hypothetical 1000 men with PCa suspicion in the development and validation cohorts according to threshold probabilities of csPCa from 1 to 100%.

## 4. Discussion

The designed BCN-RC 1 model, based on a new developed predictive model using the DRE prostate volume category, was validated in the Barcelona metropolitan area with the aim of stratifying initial suspicion of PCa to avoid unnecessary mpMRI scans. Since pre-biopsy mpMRI and guided biopsies were performed on all suspicious lesions, BCN-RC1 predicts csPCa risk even in the anterior prostate gland. The new BCN-RC 1 and BCN-RC 2, designed for selecting men for prostate biopsy after mpMRI [23], could be sequenced to establish a risk-organized model for early detection of csPCa in the Barcelona metropolitan area, where both risk calculators were validated. This risk-organized model can be especially useful in sites where access to mpMRI is limited [9,13].

The prestigious Rotterdam risk calculator was designed from the predictive models developed among the participants of the Rotterdam section of the ERSPC trial. The Rotterdam RC 3 and RC 4 calculators were designed before the spread of mpMRI to individualize the risk of PCa and high-grade PCa in biopsy-naïve men and those with prior negative prostate biopsy, respectively [33]. After the recommendation of pre-biopsy mpMRI, both risk calculators have been validated for selecting candidates for MRI [34,35]. Both Rotterdam RC 3 and RC 4 initially included the serum PSA, DRE (normal vs abnormal), hypoechoic areas in TRUS (presence vs absence), and prostate volume assessed from TRUS as predictive variables [34]. However, since 2012, it has been possible to introduce the estimated prostate volume with DRE. This modification was motivated by TRUS prostate volume usually not being available at initial PCa suspicion [19,20,21]. DRE categories were established from the TRUS prostate volume intervals of less than 30 mL, 30 to 59 mL, and 60 mL or above [34,35]. Under these conditions, Alberts et al. analysed Rotterdam RC 4 in 122 men with suspected PCa and prior negative prostate biopsy, in 26% of whom csPCa was detected. Using a 3% risk threshold for csPCa, Rotterdam RC 4 avoided 51% of mpMRI scans missing 9.7% of detected csPCa [34]. Similarly, Manners et al. analysed Rotterdam RC 3 in 200 biopsy-naïve suspected PCa men in 33.5% of whom csPCa was detected. After adjusting the risk threshold to 4%, 36.5% of mpMRI scans were avoided, missing 6% of detected csPCa [35]. Remmers et al., in a recent analysis of the current Rotterdam risk calculator without MRI, among the 206 men of the PRECISION trial MRI arm, in whom 70 csPCa were diagnosed, observed after recalibration of the model and adjustment of the risk threshold, there was 13.1% reduction in MRI scans, missing 7.1% of detected csPCa [36].

The new BCN-RC 1 includes the same clinical predictive variables as BCN-RC 2, except those derived from mpMRI. Age (years), PCa family history (yes vs no), biopsy status (naïve vs repeat), serum PSA (ng/mL), DRE (normal vs abnormal), and DRE prostate volume category (which substituted the MRI prostate volume), were found to be independent clinical predictors of csPCa in logistic regression analysis. The calibration curves showed a good agreement between predictions and the real outcome in both development and validation cohorts. The discrimination ability of csPCa was similar in both development and validation cohorts, despite their four-percentage-point difference in csPCa detection. The observed AUCs were like those reported by Alberts et al. in men with previous negative prostate biopsy [34]. Manners et al. [35] and Remmers et al. [36] did not report the discrimination ability of Rotterdam RC in their studies. In both development and validation cohorts, we observed net benefit of BCN-RC 1 over performing mpMRI for all men when from a risk threshold of csPCa between 8 to 12%. We estimated the performance of BCN-RC 1 with different sensitivities; however, with the 0.95 sensitivity of csPCa, which is in our opinion appropriate at initial PCa suspicion, the specificities in the development and validation cohort were 0.24 and 0.34, respectively, which allowed avoidance of 17.2% and 22.3% of mpMRI scans, respectively. These percentages of saved mpMRI scans were below the 51% reported with Rotterdam RC 4 and the 36.5% reported with Rotterdam RC 3, although the reported missed high-grade PCa percentages were 6% and 9.7%, respectively, compared with the up to 5% fixed for BCN-RC 1 [34,35]. In the recent Rotterdam RC 3 analysis, carried out in the PRECISION trial MRI arm, the percentage of avoided mpMRI decreased to 13.1% compared to the 36.5% reported by Manners et al., with percentages of missed csPCa of 7.1% and 6%, respectively [35,36]. We also have analysed the behaviour of BCN-RC 1 in a hypothetical 1000 men with PCa suspicion, observing that while missing 5% of detected csPCa, 410 mpMRI would be saved in the development cohort, compared to 382 in the validation cohort. Comparisons between Rotterdam RC 3 and RC 4 with BCN-RC 1 are difficult because it was developed to stratify the initial PCa suspicion of any men, whether biopsy-naïve or with previous negative prostate biopsy. The validation studies of Rotterdam RC 3 and RC 4 were performed with the based TRUS prostate volume, whereas we used the DRE prostate volume category estimated from the MRI prostate volume [19,20]. Regrettably, DRE is not routinely performed to suspect PCa [22], which is why we report the routine and complete achievement of DRE at initial PCa suspicion, assessing the prostate volume category beyond the abnormalities in the posterior prostate gland surface [20].

The web app of BCN-RC 1 was designed from a predictive model developed and externally validated in sizable cohorts of consecutive men of the same metropolitan area, with the same criteria of PCa suspicion and following the same diagnostic approach of csPCa, without limitations in age, serum PSA, or prostate volume as exist in the Rotterdam RCs. Even so, we found a 4% difference between the csPCa detection observed in development and validation cohorts, which was a stressful scenario for the predictive model. Despite this difference, the discrimination ability of the developed predictive model remained, and the rate of avoided mpMRI even increased somewhat in the validation cohort. Our study is limited by the fact that we assumed DRE accurately classifies the prostate volume category assessed from MRI. Roobol et al. made the same assumption regarding TRUS reported prostate volume due to previous evidence [19,20]. Recently, Massanova et al. confirmed a good correlation between the DRE estimated prostate volume with that assessed from MRI [37]. However, we believe that prospective analysis to define accurate MRI prostate volume intervals assessed by DRE categories is needed. Additionally, the current study takes on the limitations of predictive models that provide the specific risk of csPCa based on the landscapes in which they are developed [33]. The essential web or smartphone risk calculators need validation in each population where they will be conducted. Usually, external validations are carried out in populations with different characteristics and landscapes than those of a development cohort, and recalibrations and adjustments of csPCa risk thresholds are usually needed to obtain accurate predictions [38]. Because characteristics of populations, PCa incidence, and diagnostic approaches frequently change, making accurate real-time predictions adapted to the continuous evolution is challenging [39]. Continuous updating of risk calculators from the feedback of new cases, integrating the generation of big data, appropriate machine-learning algorithm design [40,41], and federated networks will provide the opportunity to develop future predictive models and risk calculators guaranteeing accurate and enduring overall and specific predictions [42].

Finally, we note that sequencing predictive tools are recommended for risk-orga-nised models of early detection of csPCa, first stratifying the initial PCa suspicion to reduce MRI demand and after MRI to reduce unnecessary prostate biopsies [9]. Remmers et al., after reducing 13.1% of mpMRI requests in a first stratification, observed how in a second stratification with the Rotterdam RC, MRI decreased up to 20.9% of prostate biopsies, whereas the percentage of missed csPCa increased from 7% to 8.6% [36]. Recently, the initial PCa suspicion of 2881 men has been stratified to avoid MRI scans first from the subset of men with serum PSA > 10 ng/mL and abnormal DRE [43], and thereafter with the PSA density calculated from the prostate volume estimated with DRE, resulting in a 20.3% reduction of MRI scans with 3.8% missing csPCa [21]. New combinations of appropriate tools available before and after MRI for risk-organized models of early detection of csPCa must be explored.

## 5. Conclusions

Currently, the EAU recommends risk-organized models for early detection of csPCa, stratifying the initial suspicion of PCa for avoiding MRI scans, and then stratifying men after MRI to avoid prostate biopsies. We have designed and validated BCN-RC 1 to stratify the initial PCa suspicion to reduce mpMRI demand. BCN-RC 1 includes the same clinical predictors as BCN-RC 2, except those derived from mpMRI, having substituted the MRI derived prostate volume with the DRE prostate volume category. The new BCN-RC 1 avoided between 17.2% and 22.3% mpMRI scans while missing 5% of detected csPCa, which represented 1.8% of men at risk in the development cohort and 2% in the validation cohort. We report efficient accomplishment of DRE in the early detection of csPCa for an appropriate selection for mpMRI. BCN-RC 1 is ready to be sequenced with BCN-RC 2 in a risk-organized model of csPCa.

## Figures and Tables

**Figure 1 cancers-14-05100-f001:**
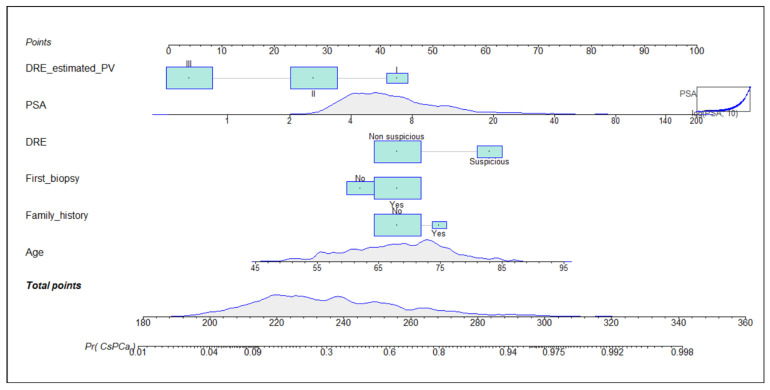
Nomogram of development model. DRE_estimated_PV = digital rectal examination prostate volume category; PSA = prostate-specific antigen; *Pr* = probability.

**Figure 2 cancers-14-05100-f002:**
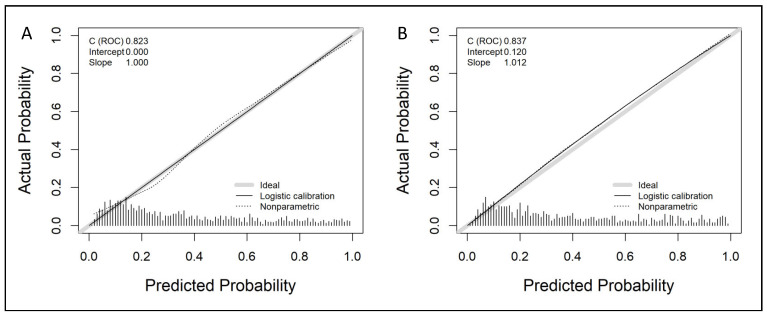
Calibration curves of predictive model in development cohort (**A**) and validation cohort (**B**).

**Figure 3 cancers-14-05100-f003:**
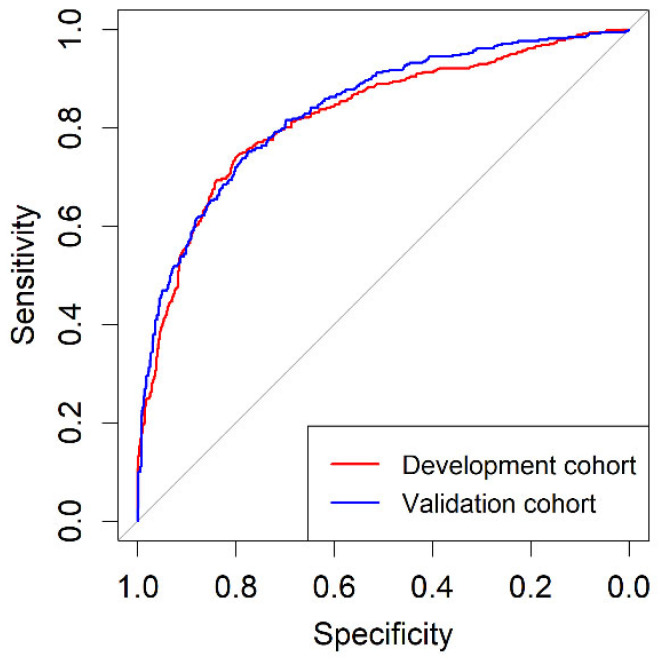
Discrimination ability represented by receiver operating characteristic curves of BCN-RC 1 in the development cohort and validation cohort.

**Figure 4 cancers-14-05100-f004:**
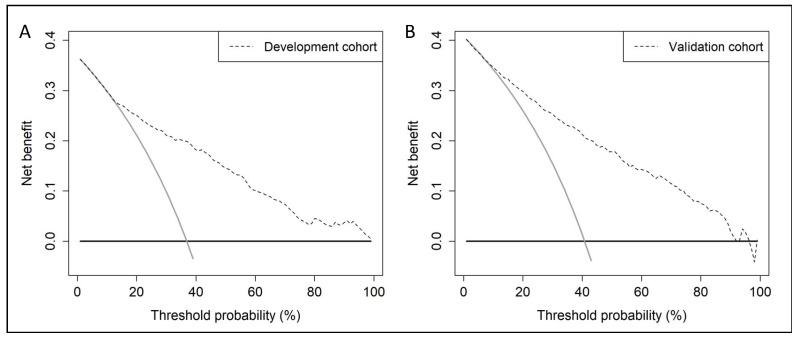
Net benefit of BCN-RC 1 over performing MRI scans represented through decision curve analysis in all men in the development cohort (**A**) and the validation cohort (**B**).

**Figure 5 cancers-14-05100-f005:**
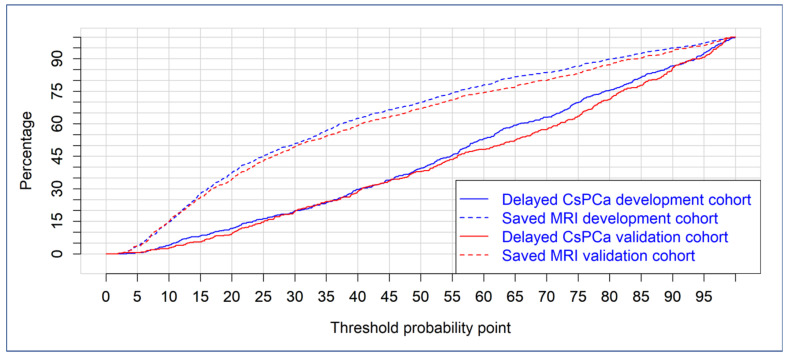
Saved mpMRI scans and missed csPCa for a continuous risk threshold of csPCa, represented through clinical utility curves.

**Table 1 cancers-14-05100-t001:** Characteristics of men making up the development and validation cohort.

Variable	Development Cohort	Validation Cohort	*p*-Value
Number of men	1486	946	-
Caucasian ethnicity, *n* (%)	1.465 (98.6)	931 (98.4)	0.738
Median age at biopsy (IQR), years	69 (62–74)	67 (61–72)	<0.001
Median serum PSA (IQR), ng/mL	6.0 (4.4–9.2)	7.4 (5.5–10.9)	<0.001
Abnormal DRE, *n* (%)	329 (22.1)	283 (29.9)	<0.001
PCa family history, *n* (%)	127 (8.5)	34 (3.6)	<0.001
Prior negative prostate biopsy, *n* (%)	388 (26.1)	293 (31.0)	=0.010
Median prostate volume (IQR), mL	55 (40–76)	55 (40–78)	=0.559
DRE-prostate volume category, *n* (%)			=0.675
I	140 (9.4)	96 (10.2)	
II	681 (45.8)	417 (44.2)	
III	665 (44.8)	431 (45.7)	
PI-RADS v.2.0, *n* (%)			<0.001
1	242 (16.3)	185 (19.6)	
2	73 (4.9)	50 (5.3)	
3	444 (29.9)	201 (21.2)	
4	450 (30.3)	391 (41.3)	
5	277 (18.6)	119 (12.6)	
PCa detection, *n* (%)	693 (46.6)	521 (55.1)	<0.001
csPCa detection, *n* (%)	548 (36.9)	386 (40.8)	=0.058
iPCa detection, *n* (%)	145 (9.8)	135 (14.3)	<0.001
csPCa detection according to PI-RADS			<0.001
<3	13 (4.1)	42 (17.9)	
3	68 (15.3)	41 (20.4)	
4	236 (52.4)	203 (51.9)	
5	231 (83.4)	100 (84.0)	

IQR = interquartile range; *n* = number; PSA = prostatic specific antigen; DRE = digital rectal examination; PI-RADS = Prostate Imaging Reporting and Data System; PCa = prostate cancer; csPCa = clinically significant PCa; iPCa = insignificant PCa; I = up to 30 mL; II = between 30 and 59 mL; III = 60 mL or above.

**Table 2 cancers-14-05100-t002:** Odds ratios and 95% confidence intervals for csPCa of independent predictive values in univariate and multivariate analysis.

Predictive Variable	Univariate OR (95% CI)	*p*-Value	Multivariate OR (95% CI)	*p*-Value
Age at biopsy, years	1.08 (1.06–1.09)	<0.001	1.08 (1.06–1.10)	<0.001
Median log serum PSA, ng/mL	8.03 (5.31–12.14)	<0.001	12.96 (7.69–21.84)	<0.001
Abnormal DRE, yes vs. no	4.51 (3.48–5.84)	<0.001	3.19 (2.34–4.34)	<0.001
PCa family history, yes vs. no	1.77 (1.23–2.56)	=0.002	1.69 (1.06–2.68)	=0.026
Prior negative prostate biopsy, yes vs. no	0.68 (0.53–0.87)	=0.002	0.63 (046–0.85)	=0.003
DRE-prostate volume category, II vs. I	0.37 (0.25–0.55)	<0.001	0.35 (0.22–0.55)	<0.001
DRE-prostate volume category, III vs. I	0.11 (0.07–0.16)	<0.001	0.07 (0.04–0.12)	<0.001

O.R. = odd ratio; C.I. = confidence interval; PCa =prostate cancer; DRE =digital rectal examination; PSA =prostate-specific antigen; I = < 30 mL; II = 30–59 mL; III = > 60 mL.

**Table 3 cancers-14-05100-t003:** Specificities of developed predictive model corresponding to sensitivities of 0.80, 0.85, 0.90, and 0.95 in the development and validation cohorts.

Sensitivity	Development Cohort		Validation Cohort		*p* Value
	Specificy (95% CI)	Threshold (%)	Specificy (95% CI)	Threshold (%)	
0.80	0.70 (0.68–0.72)	30.8	0.70 (0.67–0.73)	30.3	0.927
0.85	0.59 (0.56–0.61)	23.4	0.63 (0.59–0.66)	25.1	0.187
0.90	0.45 (0.43–0.48)	17.2	0.53 (0.49–0.56)	30.3	<0.001
0.95	0.24 (0.22–0.26)	11.1	0.34 (0.31–0.37)	13.3	<0.001

CI = confidence interval.

**Table 4 cancers-14-05100-t004:** Performance of developed predictive model of csPCa in development and validation cohorts from the 95% sensitivity threshold of csPCa of the development cohort.

Parameter	Development Cohort	Validation Cohort
Sensitivity, number (%)	520/548 (95.0)	367/386 (95.0)
Specificity, number (%)	228/938 (24.3)	192/560 34.3)
Positive predictive value, number (%)	520/1230 (42.3)	367/737 (49.8)
Negative predictive value, number (%)	228/256 (89.1)	192/209 (91.9)
Accuracy, number (%)	748/1486 (50.3)	559/946 (59.1)
Avoided mpMRI exams, number (%)	256/1486 (17.2)	211/946 (22.3)
Missed csPCa, number (%)	28/548 (5.0)	19/386 (5.0)
Odds ratio (95% confidence interval)	6.19 (4.06–9.43)	9.92 (6.06–16.24)

MpMRI = multiparametric magnetic resonance imaging.

**Table 5 cancers-14-05100-t005:** Number of missing csPCa and mpMRI scans saved for different thresholds in a hypothetical 1000 cases of PCa suspicion in development and validation cohorts.

Threshold Probability	Development Cohort		Validation Cohort	
	Missed csPCa	Saved mpMRI	Missed csPCa	Saved mpMRI
1	0	0	0	0
2	0	1	1	2
3	0	7	2	6
4	1	20	2	17
5	2	38	2	35
6	3	55	3	49
7	7	81	6	74
8	8	97	7	106
9	11	124	10	127
10	15	146	11	150
11	19	170	14	177
12	24	197	16	195
13	27	224	20	217
14	29	247	22	238
15	30	278	22	259
16	32	295	27	280
17	36	318	31	302
18	39	336	34	317
19	40	353	35	326
20	43	374	38	344
21	46	392	44	369
22	52	410	49	379
23	54	424	52	395
24	58	439	56	418
25	60	451	59	428
26	63	466	64	442
27	65	476	70	456
28	67	491	73	468
29	68	497	74	478
30	73	507	79	494
35	87	569	98	543
40	110	625	117	592
45	125	664	137	631
50	145	699	154	670
55	168	742	178	709
60	196	779	197	743
65	219	817	214	768
70	232	836	234	800
75	257	865	259	832
80	278	898	291	872
85	300	925	317	904
90	320	950	350	938
95	342	973	370	961
100	369	1000	408	1000

mpMRI = multiparametric magnetic resonance imaging; csPCa = clinically significant prostate cancer.

## Data Availability

The data presented in this study are available on request from the corresponding author.

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
