# Peer review of "A Clinically Significant Prostate Cancer Predictive Model Using Digital Rectal Examination Prostate Volume Category to Stratify Initial Prostate Cancer Suspicion and Reduce Magnetic Resonance Imaging Demand"

_cancers, 2022, doi:10.3390/cancers14205100_

Round 1

Reviewer 1 Report

Congrats to the authors for this interesting manuscript.

Herein are my few suggestions

Please, specify the name of the academic Institution where the study was conducted

It is not clear why the authors aimed to include DRE estimated prostate volume which is a poorly accurate and highly physician-dependent tool.

Please, better specify how DRE-prostate volume was defined.

Author Response

Reviewer 1.

I would like to recognize the effort of reviewers to improve articles.

Congrats to the authors for this interesting manuscript.

Thank you for this comment.

Herein are my few suggestions

Please, specify the name of the academic Institution where the study was conducted

Thank, we have included the name of the participant institutions in the study. Line 117 (Vall d´Hebron Hospital) participated in the development cohort recruitment. Lines 127-128 (Parc de Salut Mar and Germans Trias i Pujol Hospital) participated in the validation cohort recruitment.

It is not clear why the authors aimed to include DRE estimated prostate volume which is a poorly accurate and highly physician-dependent tool.

We aimed to include DRE prostate volume estimation because DRE is the only way to know this predictor of csPCa when prostate cancer suspicion is made. At that time, we only know PSA elevation and/or abnormal DRE. To assess prostate volume is necessary to perform a TRUS, which is not the current routine practice. I agree that DRE is poorly accurate and highly physician dependent tool. However, some studies confirm that it is appropriate to predict with high concordance determinate ranges of prostate volume, as Roobol et al. demonstrated incorporating DRE estimation prostate volume in the Rotterdam PCa risk calculator.

These comments are made in the introduction, lines 89-102.

Please, better specify how DRE-prostate volume was defined.

Prostate volume category was defined according to the stablished ranges of <30 mL, 30-59 mL, and >60 mL, from the prostate volume reported by MRI, according to the references 19, 20. Lines 138-140. It´s true that these studies established these ranges from the prostate volume assessed by TRUS. We recognize that low evidence exists regarding DRE prostate volume category and MRI-prostate volume, as it is commented in lines 434-349: “Recently, Massanova et al. have confirmed a good correlation between the DRE-estimated prostate volume with that assessed from MRI [38]. However, we believe that prospective analysis to define the accurate MRI-prostate volume intervals assessed by DRE categories is needed”.

Thank you very much

Juan Morote

Reviewer 2 Report

Dear Authors, I read with interest your manuscript that it is well written. The topic is very actual in the current era where a even higher number of patients required access to the medical facilities upfront to a reduced economical resources. 

In my opinion the methodology is solid and the validation of your results is well conducted.

My only suggestion is to introduce into the discussion the possibility to develop new models by using artificial intelligence.

In the past preliminary experiences were already presented with the aim to reduce the number of unnecessary biopsy (Artificial intelligence for target prostate biopsy outcomes prediction the potential application of fuzzy logic. Prostate Cancer Prostatic Dis. 2022 Feb;25(2):359-362. doi: 10.1038/s41391-021-00441-1) and also in other urological diseases (Applications of neural networks in urology: a systematic review. Curr Opin Urol. 2020 Nov;30(6):788-807. doi: 10.1097/MOU.0000000000000814).

I think that these new tools can open the doors to more sophisticated and precise predictive models.

Author Response

Reviewer 2

I would like to recognize the effort of reviewers to improve articles

Dear Authors, I read with interest your manuscript that it is well written. The topic is very actual in the current era where an even higher number of patients required access to the medical facilities upfront to a reduced economical resource. In my opinion the methodology is solid, and the validation of your results is well conducted. Thank you.

My only suggestion is to introduce into the discussion the possibility to develop new models by using artificial intelligence. In the past preliminary experiences were already presented with the aim to reduce the number of unnecessary biopsy (Artificial intelligence for target prostate biopsy outcomes prediction the potential application of fuzzy logic. Prostate Cancer Prostatic Dis. 2022 Feb;25(2):359-362. doi: 10.1038/s41391-021-00441-1) and also in other urological diseases (Applications of neural networks in urology: a systematic review. Curr Opin Urol. 2020 Nov;30(6):788-807. doi: 10.1097/MOU.0000000000000814).

I agree with the suggestion, which is in line of the comment made in the discussion, were machine learning algorithms and federated networks are recognize as important tools to develop the future risk calculators. Additionally, I have added these two important articles as references 41 and 42.  Lines 355-371. “Because characteristics of populations, PCa incidence, and diagnostic approaches frequently change, accurate real-time predictions adapted to the continuous evolution are challenging [40]. Continuous updating of risk calculators from the feedback of new cases, integrating the generation of big data, appropriate machine learning algorithms design [41,42], and federated network, will provide the opportunity to develop future predictive models and risk calculators guaranteeing accurate and enduring overall and specific predictions [43]”. 

I think that these new tools can open the doors to more sophisticated and precise predictive models.                                                                                                                           I agree with you.

Thank you very much,                                                                                                             Juan Morote
